# Protocols of Anesthesia Management in Parturients with SARS-CoV-2 Infection

**DOI:** 10.3390/healthcare10030520

**Published:** 2022-03-12

**Authors:** Antonio Coviello, Maria Vargas, Annachiara Marra, Ludovica Golino, Gabriele Saccone, Carmine Iacovazzo, Maria Grazia Frigo, Andrea Tognù, Marilena Ianniello, Pasquale Buonanno, Giuseppe Servillo

**Affiliations:** 1Department of Neurosciences, Reproductive and Odontostomatological Sciences, University of Naples “Federico II”, 80100 Naples, Italy; vargas.maria82@gmail.com (M.V.); dottmarraannachiara@gmail.com (A.M.); gabriele.saccone.1990@gmail.com (G.S.); dott.iacovazzo@gmail.com (C.I.); marilena.ianniello@gmail.com (M.I.); pasqual3.buonanno@gmail.com (P.B.); dainuctar.aou@unina.it (G.S.); 2Department of Anesthesiology and Intensive Care, San Giovanni di Dio Hospital, 80027 Naples, Italy; ludovica.golino00@gmail.com; 3Department of Anesthesia and Resuscitation in Obstetrics, San Giovanni Calibita Fatebenefratelli Hospital, 39, 00186 Rome, Italy; mariagrazia.frigo@gmail.com; 4Department of Anesthesiology and Intensive Care Medicine, Istituto Ortopedico Rizzoli IRCCS, 40136 Bologna, Italy; andrea.tognu@ior.it

**Keywords:** anesthesia, cesarean delivery, COVID-19, delivery, epidural analgesia, fetus, labor, neuraxial anesthesia, pregnancy, pneumonia, safety, SARS-CoV-2, spinal anesthesia, 2019-nCOV

## Abstract

Background: Our hospital became a referral center for COVID-19-positive obstetric patients from 1 May 2020. The aim of our study is to illustrate our management protocols for COVID-19-positive obstetric patients, to maintain safety standards for patients and healthcare workers. Methods: Women who underwent vaginal or operative delivery and induced or spontaneous abortion with a SARS-CoV-2-positive nasopharyngeal swab using real-time PCR (RT-PCR) were included in the study. Severity and onset of new symptoms were carefully monitored in the postoperative period. All the healthcare workers received a nasopharyngeal swab for SARS-CoV-2 using RT-PCR serially every five days. Results: We included 152 parturients with COVID-19 infection. None of the included women had general anesthesia, an increase of severe symptoms or onset of new symptoms. The RT-PCR test was “negative” for the healthcare workers. Conclusions: In our study, neuraxial anesthesia for parturients’ management with SARS-CoV-2 infection has been proven to be safe for patients and healthcare workers. Neuraxial anesthesia decreases aerosolization during preoxygenation, face-mask ventilation, endotracheal intubation, oral or tracheal suctioning and extubation. This anesthesia management protocol can be generalizable.

## 1. Introduction

In December 2019, a novel coronavirus was identified in Wuhan, China [1]. In the subsequent weeks, the infection rapidly spread across China and worldwide [2]. On 12 February, the World Health Organization (WHO) named the disease as Coronavirus Disease 2019 (COVID-19) [3].

COVID-19 patient management faces enormous challenges as we try to guarantee the safety of the patient and to ensure the safety and health of the operators involved in the care process. For this reason, it is necessary for the operators to properly wear personal protective equipment. Whenever possible, patients are required to wear surgical masks. With reference to the anesthesia management of COVID-19 patients, it is important to minimize aerosol-generating procedures, i.e., procedures that produce an aerosol of respiratory secretions, such as: intubation, extubation or related procedures, manual ventilation, open suctioning and non-invasive ventilation. Therefore, local anesthesia and regional anesthesia are preferred whenever possible to avoid invasive airway management, e.g., tracheal intubation.

Our hospital became a regional referral center for COVID-19 parturients in May 2020. 

The objective of this study is to validate the safety and effectiveness of anesthesia management protocols for parturients with COVID-19 infection, for both patients and providers (surgeons, airway managers, airway operators, intubators).

## 2. Materials and Methods

This is a cohort study of parturients with SARS-CoV-2 infection referred to the University of Naples, Federico II (Naples, Italy), from 1 May to 30 November 2020. Ethical approval for this study (Ethical Committee No. 145/20) was provided by the Ethical Committee, University of Naples, Federico II, Italy (Prof Domenico Del Forno), on 3 April 2020. Women who underwent vaginal or operative delivery and induced or spontaneous abortion with a SARS-CoV-2-positive nasopharyngeal swab using real-time PCR (RT-PCR) were included in the study. Written consent of the patients was required for inclusion in the study.

In our center, a COVID-19-dedicated pathway to obstetric patients was set up, including a delivery room, an operating room and a hospital ward. Each room was equipped with a clean pathway, with a room used for dressing the operators, and a dirty pathway, with a room used for undressing and sanitizing the operators.

All women admitted received a nasopharyngeal swab for SARS-CoV-2 using RT-PCR. Patients who tested positive and those who arrived with a “positive” RT-PCR test were placed in a dedicated ward.

All the providers used the following personal protective equipment kit (PPE): waterproof suit, double boots, double gloves, protective goggles, Filtering Facepiece 3 (FFP3) mask and visor [4,5]. Patients were required to wear surgical masks. 

Patients were divided into four groups:Group 1: patients scheduled for planned cesarean delivery (78 patients)Group 2: patients who underwent an emergent or urgent cesarean delivery (10 patients)Group 3: patients who underwent vaginal delivery (44 patients)Group 4: patients who underwent spontaneous or induced abortion (20 patients)

### 2.1. Interventions

Perioperative management was performed according to institutional standards. Routine monitoring during the entire perioperative period included pulse oximetry (SpO2), five-lead ECG and non-invasive pressure. 

#### 2.1.1. Techniques

##### Group 1: Planned Cesarean Delivery

The double-space combined spinal-epidural (CSE) technique was performed with the patients in a sitting position. The epidural space was identified at the L2–L3 intervertebral level with an 18-gauge Tuohy needle using the loss of resistance with saline technique. Subsequently, a 27-gauge Whitacre spinal needle was advanced at the L3–L4 intervertebral level through an introducer until cerebrospinal fluid (CSF) was obtained and bupivacaine 0.5% 10 mg was injected. The epidural catheter was inserted 5 cm into the epidural space. The test dose was performed with lidocaine 2% 3 mL and epinephrine 15 mcg. 

Postoperative analgesia was provided with an epidural bolus of ropivacaine 0.1% 10 mL, followed by a Patient Intermittent Epidural Boluses/Patient Controlled Epidural Analgesia (PIEB/PCEA) pump with programmed boluses of ropivacaine 0.1% (8 mL per hour) and the possibility of boluses at the patient’s request (5 mL per hour, with safety lock for the next 60 min).

##### Group 2: Emergent or Urgent Cesarean Delivery

Spinal anesthesia was performed with the patient in the sitting position. In asepsis, a 27-gauge Whitacre spinal needle was advanced at the L3–L4 intervertebral level through an introducer until CSF was obtained and bupivacaine 0.5% 10 mg and sufentanil 5 mcg were injected. 

##### Group 3: Vaginal Delivery

With the patient in the sitting position, the epidural space was identified at the L2–L3 intervertebral level with an 18-gauge Tuohy needle using the loss of resistance with saline technique. The epidural catheter was inserted 5 cm into the epidural space. The mother decided whether or not to use the catheter for labor analgesia. The epidural catheter was used for postnatal analgesia in case of episiotomy or perineal lacerations. 

##### Group 4: Spontaneous or Induced Abortion

Spinal anesthesia was performed with the patient in the sitting position. In asepsis, a 27-gauge Whitacre spinal needle was advanced at the L3–L4 intervertebral level through an introducer until CSF was obtained and bupivacaine 0.5% 10 mg and sufentanil 5 mcg were injected. 

Incidence of intraoperative adverse effects induced by neuraxial anesthesia such as nausea, vomiting, hypotension and bradycardia was reported.

In the postoperative period, patients were contacted every 6 h by video-call for the first five days. Visual Analogic Scale (VAS), presence of adverse effects such as nausea or vomiting or pruritus, severity and onset of news symptoms were carefully monitored. Pharmacological therapy was targeted according to patients’ responses. 

All the healthcare workers involved received a nasopharyngeal swab for SARS-CoV-2 using RT-PCR every five days.

Primary outcomes were:Patients’ peri-operative safety and effectiveness, observing vital signs and anesthesia quality.

Secondary outcomes were:Providers’ peri-operative safety was assessed by the nasopharyngeal swab for SARS-CoV-2 result every five days by RT-PCR.General anesthesia rate.

Our aim was to perform loco-regional anesthesia in all patients. Patients with critical respiratory symptoms treated with NIV or HFNC were supported only with oxygen therapy (FiO_2_ 60%) during cesarean section. Opioid-free general anesthesia was the contingency plan for COVID-19 parturients when regional anesthesia and analgesia were not possible. 

### 2.2. Statistical Analysis

Categorical variables were expressed in percentages and compared with the Chi-square test. Continuous variables were reported as mean ± standard deviation and compared with the Student’s *t*-test for unpaired samples. Statistical significance was set with a *p*-value of 0.05. The confidence interval (CI) was 95%. Statistical analysis was performed using SPSS version 20.0 (IBM Inc., Armonk, NY, USA).

## 3. Results

During the study period, 152 parturients with COVID-19 infection were included in the study (Appendix A Figure A1: Flow chart). During the study period, there were about 2500 COVID-19-negative parturients. We have experienced a significant increase in the number of pregnant patients affected by COVID-19 since September 2020 (Figure 1). Cesarean section rate was higher than usual (47%).

Most of our patients showed infection symptoms—only 12 parturients with COVID-19 infection were asymptomatic. Headache, cough, nausea and vomiting were the most presented symptoms (Table 1). Among the symptomatic parturients, 112 patients had mild symptoms, 21 patients had severe symptoms (high fever above 39 °C, persistent pain or pressure in the chest, new-onset confusion status) and 7 patients had critical symptoms (dyspnea, shortness of breath, hypoxia, blue lips or face). No woman enrolled in the study had an increase in severity or onset of new symptoms related to COVID-19 infection in the first five days of the postoperative period.

We have handled 4 different groups of patients: 78 patients who had a planned cesarean delivery (Group 1: 51.3%)—among these we reported 2 hysterectomies (a patient on their fifth pregnancy with 4 previous caesarean sections and a patient with placenta accreta), 10 patients who underwent emergent or urgent cesarean delivery (Group 2: 6.6%), 44 who underwent vaginal delivery (Group 3: 29%), none of whom had an emergent or urgent caesarean section but 4 underwent operative vaginal deliveries with a vacuum extractor, and 20 patients who had a spontaneous or induced abortion (Group 4: 13.1%) (Figure 2).

Patients with severe or critical symptoms underwent a programmed cesarean section and received preoperative oxygen therapy, respectively with nasal cannula or venti mask, and high-flow nasal cannula (HFNC) or noninvasive ventilation (NIV). We noted an improvement (up to 10% more than the measurement before extraction) in peripheral oxygen saturation (SpO2) in the post-delivery period compared with the pre-delivery period in all patients with severe or critical symptoms (*p* < 0.05) (Table 2). In the postoperative period, the fraction of inspired oxygen (FiO2) was gradually reduced in all patients.

Table 3 shows the incidence rate of the intraoperative symptoms in pregnant patients undergoing cesarean section.

We reported two maternal admissions to the intensive care unit (ICU), one due to hemolysis, elevated liver enzymes and a low platelet count syndrome (HELLP syndrome), and one for postpartum hemorrhage.

None of the 152 included women had general anesthesia or were admitted in intensive care for problems related to COVID-19.

During the study period, no health workers involved tested positive for the nasopharyngeal swab for SARS-CoV-2 using RT-PCR (Table 4).

## 4. Discussion

In our study, neuraxial anesthesia for the management of parturients with SARS-CoV-2 infection has been proven to be safe for patients and healthcare workers.

Parturients undergo important anatomical and physiological changes during pregnancy. These changes affect every organ system of the pregnant body through both biochemical and mechanical pathways, in response to the development of the fetus [6]. Therefore, parturients are at increased risk for severe illness from respiratory infections [7].

Data on COVID-19 during pregnancy are growing worldwide [8,9,10]. 

The past, years of experience about two other coronaviruses, the Severe Acute Respiratory Syndrome Coronavirus (SARS-CoV) and the Middle East Respiratory Syndrome Coronavirus (MERS-CoV), suggest that parturients are more susceptible to adverse outcomes, including the need for endotracheal intubation and admission to an intensive care unit [10,11,12,13].

Goals for obstetric anesthesia in COVID-19-positive parturients must include both patients’ comfort and healthcare workers’ safety. 

In our study, all admissions underwent a molecular swab for SARS-CoV-2 and positive patients were placed in a different pathway from other patients, including dedicated delivery rooms and dedicated operating rooms. Similarly, patients who arrived from other hospitals with positive swabs were addressed to the COVID-19 pathway. All patients who came to our obstetric emergency rooms were screened for suspicious contacts and symptoms such as fever, cough, sore throat and new loss of taste or smell shown in the previous two weeks. Aerosol-generating procedures such as endotracheal intubation and extubation are dangerous, and the involved clinicians must apply airborne precautions; however, some experts recommended airborne precautions for all patients who undergo surgery, since electrocautery and open surgery can generate aerosols of body fluids [14,15,16]. Appropriate PPE includes N95/FP3 or other respirators that offer a higher level of protection, eye protection (goggles, face shield, or full-face PAPR (powered air-purifying respirator)), gloves, a water-resistant gown, shoe covers and disposable operating room caps and beard covers [17,18,19]. Healthcare workers should pay special attention to the appropriate sequence of both putting on (donning) and taking off (doffing) PPE to avoid contamination [20,21]. 

COVID-19 is not a contraindication to neuraxial anesthesia in delivery; moreover, it may avoid the necessity of airway management and prevent the risk of virus spread [22]. 

From our point of view, there has been an increase in caesarean sections, particularly in the early stages of the pandemic. In those phases, the management of pregnant COVID-19 patients was complicated, and the caesarean section allowed a safer and quicker birth and reduced the exposure of the operators to COVID-19. 

The neuraxial technique is not contraindicated even in preeclamptic or eclamptic patients, like our patient who was subsequently admitted to the ICU for HELLP syndrome and posterior reversible encephalopathy syndrome (PRES) [23]. Patients with COVID-19 often receive venous thromboembolism prophylaxis. An anticoagulation regimen should be carefully considered so that neuraxial anesthesia techniques may be used [24,25]. Thrombocytopenia has been reported in both pregnant and non-pregnant patients with COVID-19 [26]. The needle for spinal anesthesia must be inserted in a mid-to-low lumbar intervertebral space. 

The prophylactic administration of ondansetron and crystalloids (co-loading) can stabilize pregnant hemodynamics and reduce the incidence of adverse effects induced by neuraxial anesthesia, such as nausea, vomiting, hypotension and bradycardia [27,28]. The test dose was avoided in Group 3 because each local anesthetic epidural dose used for analgesia can effectively replace it during labor and vaginal delivery [29]. 

Sedation should be avoided if possible and—whether supplemental oxygen is required—the lowest possible flows to maintain oxygenation should be used. Patients should always wear a surgical mask. Patients with critical respiratory symptoms treated with NIV or HFNC were supported only with oxygen therapy (FiO_2_ 60%) during cesarean section. We noticed an improvement in respiratory dynamics, SpO2 (up to 10% more than the measurement before extraction) and PaO_2_/FiO_2_ ratio in patients with severe and critical symptoms after delivery [30]. The improvement in respiratory dynamics is probably due to the lung basal segments’ recruitment, collapsed from the presence of the fetus.

Since many patients could be asymptomatic upon admission, a pre-admission triage system to screen parturients for COVID-19 positivity is crucial, as is the creation of separate facilities consisting of a delivery room, operative room (OR) and recovery area, with adequate space for donning and doffing of PPE by the healthcare workers. 

Despite the diminished visibility due to fogging, and the diminished sensibility due to triple layers of gloves, we did not experience failed attempts with standard devices. 

Tele-consultation and video-calls could help clinicians in preparing specific materials and drugs, avoiding unnecessary contacts [31].

We suggest that parturients are not mainly affected by respiratory complications of COVID-19 when compared with the outcomes described in the general population. However, the small number of reported cases during pregnancy does not allow us to affirm that COVID-19 is more aggressive during pregnancy.

## 5. Conclusions

In summary, the growing number of COVID-19-positive parturients is challenging medical systems and anesthesiologic practices. Anesthesia management of COVID-19-positive parturients must include both safe care for patients and special precautions for healthcare workers. The use of regional anesthesia, strongly recommended for labor and caesarean section, could be harder due to the need to ensure adequate PPE (contact/droplet protection and airborne protection), but it is safe and effective for women and healthcare workers.

Furthermore, anticipation of emergencies is essential, and all strategies should be applied to avoid general anesthesia and unexpected hypoxia.

In our study, we planned neuraxial anesthesia for all patients in order to decrease aerosolization during preoxygenation, face-mask ventilation, endotracheal intubation, oral or tracheal suctioning and extubation.

## Figures and Tables

**Figure 1 healthcare-10-00520-f001:**
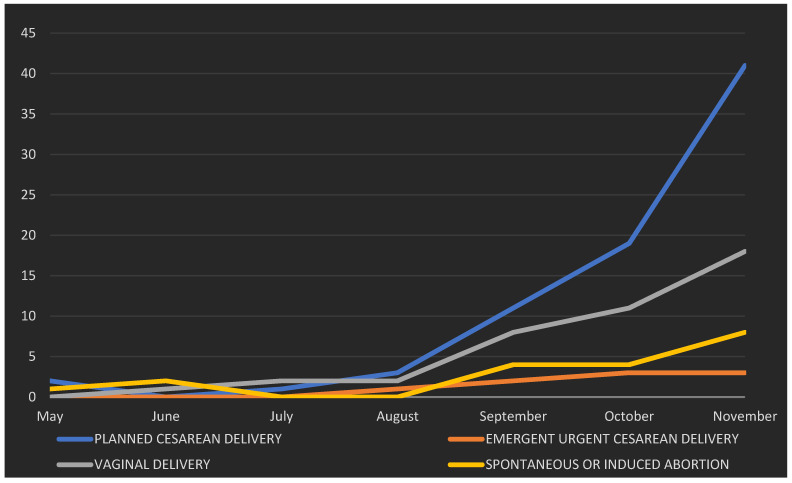
Temporal distribution of cases in the period from May to November 2020.

**Figure 2 healthcare-10-00520-f002:**
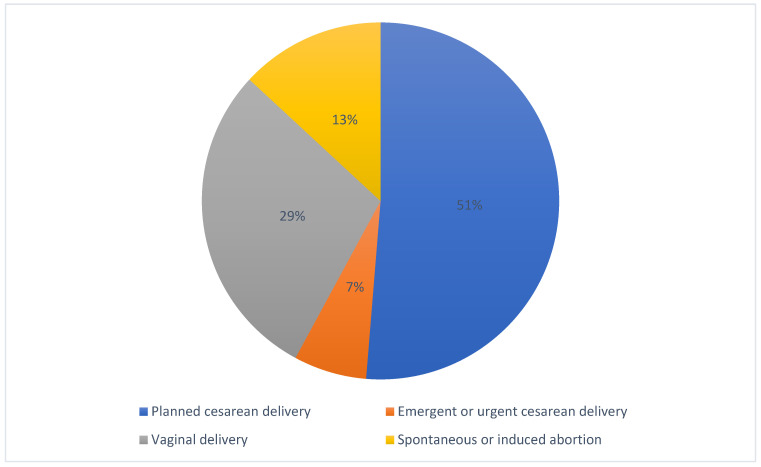
Characteristics of the included women.

**Table 1 healthcare-10-00520-t001:** COVID-19-positive pregnant patients’ symptoms in the preoperative period.

Symptoms	Number of Patients (Proportion, %)
Fever	28 (20%)
Cough	54 (38.6%)
Loss of taste/smell	21 (15%)
Malaise	35 (25%)
Dyspnea	7 (5%)
Myalgia	19 (13.6%)
Sore throat	31 (22.1%)
Headache	56 (40%)
Nausea/Vomiting	42 (30%)
Diarrhea	10 (7.2%)
Fatigue	12 (8.6%)
Chills	7 (5%)
Runny nose	13 (9.3%)

**Table 2 healthcare-10-00520-t002:** Data relating to pregnant patients (Pts) with critical or severe respiratory symptoms.

Title	Pregnant Pts with Severe Symptoms	Pregnant Pts with Critical Symptoms	*p*-Value
SpO2 pre-extraction (%)	91.3 ± 2.3	85.7 ± 1.7	<**0.05**
post-extraction (%)	95.9 ± 2.3	94.1 ± 2.0	
Oxygen therapy (type of) NIV (No. of Pts with)	0 (0.0%)	4 (57.1%)	<**0.05**
HFNC (No. of Pts with)	0 (0.0%)	3 (42.9%)	
NC (No. of Pts with)	7 (33.3%)	0 (0.0%)	
Venturi mask (No. of Pts with)	14 (66.7%)	0 (0.0%)	

Values are mean, standard deviation (SD) or number of patients (proportion, %). Patients (Pts), non-invasive ventilation (NIV), high-flow nasal cannula (HFNC), nasal cannula (NC). Boldface data are statistically significant.

**Table 3 healthcare-10-00520-t003:** Intraoperative symptoms in pregnant patients undergoing cesarean section.

Intraoperative Symptoms	Number of Patients (Proportion, %)
Nausea/Vomiting	4 (3.4%)
Hypotension	6 (6.8%)
Bradycardia	3 (3.4%)

Number of patients (proportion, %).

**Table 4 healthcare-10-00520-t004:** Monitoring of 60 healthcare workers using SARS-CoV-2 nasopharyngeal swab performed every 5 days from May to November 2020: monthly report.

Period	No. of Positive Swabs
May 2020	0 (0.0%)
June 2020	0 (0.0%)
July 2020	0 (0.0%)
August 2020	0 (0.0%)
September 2020	0 (0.0%)
October 2020	0 (0.0%)
November 2020	0 (0.0%)

Number of patients (proportion, %).

## Data Availability

All data generated or analyzed during this study are included in this published article.

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
