# Peer review of "Protocols of Anesthesia Management in Parturients with SARS-CoV-2 Infection"

_healthcare, 2022, doi:10.3390/healthcare10030520_

Round 1

Reviewer 1 Report

I have carefully read the manuscript by Coviello et al who report their experience with pregnant women affected by SarsCov2.
The management described is basically a management of organizational clinical prophylaxis against the potential infection of the operators and the contamination of the paths and spaces.
Compared to a control group not affected by SarsCov2 infection, what changes from the point of view of the anesthetic strategy?
I believe nothing. Therefore, the description of the anesthetic conduct must be absolutely reduced and schematized.
Instead, what is very important to clarify is precisely the data on SarsCov2 disease that affects the group of patients observed.
In fact, it appears that 140 out of 152 patients have a symptomatic disease and that 21 have severe and 7 even critical disorders.
Now the worrying symptoms of SarsCov2 disease are essentially respiratory ones and even if there has been much talk of silent hypoxia in Covid-19, however, the data of dyspnea present in only 5% of pregnant women must be specified.
How many women had preoperative, intraoperative and postoperative oxygen therapy?
In critical cases, has the advantageous use of HFNCs been considered?
The use of the ICU also occurred for problems not inherent to Covid-19.
It seems that the only concern was to protect operators, while a description of measures aimed at making delivery safe even in critical conditions such as Covid-19 would be desirable.
The population described appears to be more a population with mild disease, where except for PPE, the rest is the classic treatment of full-term pregnancy.

I believe that the work is of some interest, but that the anesthetic part (conduct and drugs) must be reduced and perhaps put in a table / scheme. More data should be reported regarding pre-during and post anesthesia oxygenation.

Author Response

Healthcare-1574741

Protocols of Anesthesiologic Management in pregnant women with SARS-COV-2 infection

Dear Editor,

thank you for the time spent in reviewing our manuscript and for the opportunity to re-revise it. We have carefully considered all the comments by reviewers and we have addressed them in a point-by-point way. Please, find a table below with a list of our replies and the changes applied to the paper. We do believe that it’s now much improved and we hope you will find it suitable for publication in your journal.

We look forward to hearing from you.

Best regards

The authors                                                  

Comment

Reply

Line

Reviewer #1

I have carefully read the manuscript by Coviello et al who report their experience with pregnant women affected by SarsCov2.

The management described is basically a management of organizational clinical prophylaxis against the potential infection of the operators and the contamination of the paths and spaces.

Compared to a control group not affected by SarsCov2 infection, what changes from the point of view of the anesthetic strategy? I believe nothing. Therefore, the description of the anesthetic conduct must be absolutely reduced and schematized.

Thank you for this remark. Our anesthetic strategy compared to a control group not affected by SarsCov2 infection changes in the vaginal delivery group and in the spontaneous or induced abortion group. The epidural catheter was positioned before usual in all women although it was not used for childbirth analgesia. We normally performed sedation in spontaneous or induced abortion. The description of the anesthetic conduct has been edited as suggested.

From 217 to 277

Instead, what is very important to clarify is precisely the data on SarsCov2 disease that affects the group of patients observed.

In fact, it appears that 140 out of 152 patients have a symptomatic disease and that 21 have severe and 7 even critical disorders.

Now the worrying symptoms of SarsCov2 disease are essentially respiratory ones and even if there has been much talk of silent hypoxia in Covid-19, however, the data of dyspnea present in only 5% of pregnant women must be specified.

How many women had preoperative, intraoperative and postoperative oxygen therapy? In critical cases, has the advantageous use of HFNCs been considered?

Thank you, the text has been edited as suggested.

274-275

293-300

From 310 to 315

From 379 to 384

Table 2

The use of the ICU also occurred for problems not inherent to Covid-19.
It seems that the only concern was to protect operators, while a description of measures aimed at making delivery safe even in critical conditions such as Covid-19 would be desirable.
The population described appears to be more a population with mild disease, where except for PPE, the rest is the classic treatment of full-term pregnancy.

I believe that the work is of some interest, but that the anesthetic part (conduct and drugs) must be reduced and perhaps put in a table / scheme. More data should be reported regarding pre-during and post anesthesia oxygenation.

Thank you for this interesting remarks. We reported two maternal admissions to Intensive Care Unit (ICU) , one due to Hemolysis, Elevated Liver enzymes, and a Low platelet count syndrome (HELLP syndrome), another for postpartum hemorrhage. None of the 152 included women had general anesthesia or was admitted in intensive care for problems related to Covid-19.

In our opinion, anesthesia management proposed could be the measure aimed at making delivery safe even in critical conditions such as Covid-19.

The text has been edited as suggested.

From 317 to 319

From 217 to 277

From 293 to 300

From 374 to 396

Table 2

Table 3

Table 4

Reviewer 2 Report

I thank the editor for the opportunity to review this interesting article by my colleagues from Naples; the article highlights a management protocol for giving birth patients on a single center and I find the study interesting. Unfortunately, in my opinion, there are a couple of major concerns that must be completely addressed before the article can be published.

M&M:
In the last sentence of the introduction, the authors talk about the aim of the study; they speak of "safe for patients and for providers". Unfortunately, during the entire M&M session, despite the brilliant description of the methodology for childbirth, neither a safety measure for patients nor for operators is evident. How many operators got infected? How did they measure the “Safety” for a patient? How has the use of the indicated devices affected the "safety" for patients?

A part of the description of the primary and secondary outcomes is missing. Furthermore, the description of the statistical analysis performed is missing. Even if the statistic is relatively simple (it is a descriptive statistic of frequency), it is necessary to describe what has been performed as a type of calculation, with which program, etc ...

Ethics Committee and informed consent: the Ethics Committee expressed its opinion 1 month after the first patients were enrolled; how did you proceed with the informed consent for these patients? Did they give their consent before the study was approved? It is an extremely important fact to be evaded.

Conclusions:
They are too long and are not related to the results that have been found, in particular with regard to safety which is not highlighted.

I think the work is good, but it needs to be finished properly; in particular by giving a more "scientific" structure by inserting the outcomes, the statistical part and giving a "priority" of description of the results to be described and which have been described.

Author Response

Healthcare-1574741

Protocols of Anesthesiologic Management in pregnant women with SARS-COV-2 infection

Dear Editor,

thank you for the time spent in reviewing our manuscript and for the opportunity to re-revise it. We have carefully considered all the comments by reviewers and we have addressed them in a point-by-point way. Please, find a table below with a list of our replies and the changes applied to the paper. We do believe that it’s now much improved and we hope you will find it suitable for publication in your journal.

We look forward to hearing from you.

Best regards

The authors

Comment

Reply

Line

Reviewer #2

I thank the editor for the opportunity to review this interesting article by my colleagues from Naples; the article highlights a management protocol for giving birth patients on a single center and I find the study interesting. Unfortunately, in my opinion, there are a couple of major concerns that must be completely addressed before the article can be published.

M&M:
In the last sentence of the introduction, the authors talk about the aim of the study; they speak of "safe for patients and for providers". Unfortunately, during the entire M&M session, despite the brilliant description of the methodology for childbirth, neither a safety measure for patients nor for operators is evident. How many operators got infected? How did they measure the “Safety” for a patient? How has the use of the indicated devices affected the "safety" for patients?

Thank you, the text has been edited as suggested. We added the providers safety assessment in M&M and in results.

From 217 to 277

From 310 to 323

Table 3

Table 4

A part of the description of the primary and secondary outcomes is missing. Furthermore, the description of the statistical analysis performed is missing. Even if the statistic is relatively simple (it is a descriptive statistic of frequency), it is necessary to describe what has been performed as a type of calculation, with which program, etc ...

Thank you, the text has been edited as suggested.

From 279 to 285

From 268 to 273

Ethics Committee and informed consent: the Ethics Committee expressed its opinion 1 month after the first patients were enrolled; how did you proceed with the informed consent for these patients? Did they give their consent before the study was approved? It is an extremely important fact to be evaded.

Thank you, we apologize for the mistake. Patients enrollment began in May 2020, after the Ethics Committee Approval. The text has been edited.

100 – 183 – 193

Fig 1

Conclusions:
They are too long and are not related to the results that have been found, in particular with regard to safety which is not highlighted.

I think the work is good, but it needs to be finished properly; in particular by giving a more "scientific" structure by inserting the outcomes, the statistical part and giving a "priority" of description of the results to be described and which have been described.

Thank you, the text has been edited as suggested.

From 401 to 411

Reviewer 3 Report

Manuscript Number: healthcare-1574741

Authors: Coviello et al.

Title of the manuscript: Protocols of Anesthesiologic Management ….

Minor comments:

  • Reference hospital? (referral hospital)
  • Anesthesiologic management? (anesthesia management)
  • Pregnant women? (parturients)
  • The authors capitalized letters of the words in the text where is not necessary. For example, in page 2, Cefazolin, Clindamycin, Cerebro-Spinal Fluid, Gauge Bupivacaine, Epinephrine, ….etc. Meanwhile, the terms of SpO2, ECG, NIBP, MAP, SAP, etc should be spelled out when appeared for the first time in the text.

Abstract

  • Background: “The objective of the study is to illustrate…” should be stated in the Background, instead of in the Conclusion.
  • Method: Instead of describing Inclusion method, other details should be included.
  • Results: Too simplified.
  • Conclusion: Should make a conclusion which is supported by the solid evidence from study results.

Introduction

  • Page 1, line 36: “aerosol-generating procedures” appeared in the text for the first time. Therefore, it would be better to describe this term here.
  • Page 1, line 37: “loco-regional anesthesia”? (local anesthesia and regional anesthesia)
  • Page 1, line 33, 38, etc: “operators”? (surgeons, airway managers, airway operators, intubators)
  • Page 1, line 37: “to avoid the management of the patient airway”? (to avoid invasive airway management, e.g., tracheal intubation, on the parturients)
  • Page 1, line 40: “management protocols in anesthesiology”? (protocols for anesthesia management)
  • Page 1, line 40: “The objective of this study is to validate….”? (…..the safety and effectiveness of anesthesia management….)
  • Page 2, line 45: The IRB approval letter was issued on April 3, 2020 and the study began from March 1 to November 30, 2020. So, is this study categorized as prospective or retrospective study?
  • Page 2, line 51: “A total of 152 patients were included…”? Is there any data for the recruitment rate?
  • Page 2, line 57: “All women admitted for inpatient admission..”? (..admitted as an inpatient…)
  • Page 2, line 58: “Positive patients were placed in a dedicated room, along with those who arrived already with a RT-PCR test.”? (…already with a “positive” RT-PCR test?)
  • Page 2, line 61: Regarding the role of PPE, the authors cite the reference number 4, which is a scenario-design using standard patients. Any other guidelines papers are available for citing the role of PPE for COVID-19 ?
  • Page 2, line 49: There were 152 patients tested positive using RT-PCR and included in this study. In comparison to these 152 COVID-positive obstetrical patients, how many were there COVID-negative obstetrical patients during this study period?
  • Page 2, line 64: There were 78+10 (88) subjects received Cesarean section and 44 received vaginal delivery. Is this Cesarean section rate much higher than usual?
  • In the Method section, (Page 2 to 3), the authors described all the details of the anesthesia and delivery-related issues. Are all of these details of the procedures for COVID-positive patients different from usual routines and medications (except adopting PPE)?
  • Page 2, line 95: Regarding PCEA for the group 1, the authors used ropivacaine without any other narcotics for rescue. Meanwhile, side effects of narcotics (page 3, line 99, nausea, vomiting, pruritus) were monitored every 6 hours post-operatively. Any other pain control was used postoperatively?
  • Page 3, lines 112 and 138: “Sufentanyl” should be spelled as “sufentanil”
  • Page 4, line 153, Page 5, line 169: Please define “severe and critical symptoms”
  • Page 4, line 161: “None of the 152 included women had general anesthesia.” Perhaps it is helpful to discuss later in the Discussion section, if general anesthesia is required in such situation, what would you have done?

Discussion

  • Regarding the severity and critical conditions of the COVID patients, is there any assessment to better describe 152 included patients in more details (c.f. Table 2)?
  • In those more severe or critical conditions of the subjects included in this study, is the treatment course different from those not included in this study in your hospital?
  • Page 6, line 184: Regarding the statement “women are more susceptible to adverse outcome, including the need for endotracheal intubation and admission to an intensive care unit”, any difference between those and your results?
  • Page 6, line 210: “However, cesarean delivery was used to assist mothers and fetus safely by reducing the coronavirus exposure by operators. Over time, the procedures have improved and optimized.” Please explain what have been improved and optimized?
  • Page 6, line 216: Regarding the issues of “thromboembolism prophylaxis”, “thrombocytopenia”, any data or comparison can be provided in this study?
  • Page 6, line 221: Regarding the issues of “side effects of neuraxial anesthesia such as nausea, vomiting, hypotension, and bradycardia.”, any data can be provided in this stud?
  • Page 7, line 230: Regarding “SpO2 (up to 10% more than the measurement before extraction) and PaO2/FiO2 ratio ….”, in the Methods and Results section, only NIBP was mentioned. Any A-line was implemented or only POC?
  • Page 7, line 233: Regarding “Avoid allowing Cerebro-Spinal Fluid (CSF) to drip from the needle and avoid contact with CSF. ”, was there any chance to collect CSF for PCR testing in this study?
  • Page 7, line 235: “In our experience we planned neuraxial anesthesia for all patients scheduled for elective procedure in order to decrease aerosolization during preoxygenation, face-mask ventilation, endotracheal intubation, oral or tracheal suctioning and extubation.” The emergent Cesarean section for group 2 was elective?

The main issues needed to be discussed or presented

  • How many and how severe COVID patients are identified in this 152-patient study? Are there any differences from those COVID patient without pregnancy?
  • What is contingency plan for those COVID-parturients when regional anesthesia and analgesia were not possible?
  • While suggesting the neuraxial anesthesia and analgesia is better and safer for COVID-positive parturients during the pandemic, the authors better to provide some indices to support this statement (e.g., safety parameters, efficiency indices, outcomes,..)

Author Response

Healthcare-1574741

Protocols of Anesthesiologic Management in pregnant women with SARS-COV-2 infection

Dear Editor,

thank you for the time spent in reviewing our manuscript and for the opportunity to re-revise it. We have carefully considered all the comments by reviewers and we have addressed them in a point-by-point way. Please, find a table below with a list of our replies and the changes applied to the paper. We do believe that it’s now much improved and we hope you will find it suitable for publication in your journal.

We look forward to hearing from you.

Best regards

The authors

Comment

Reply

Line

Reviewer #3               

Minor comments:

Reference hospital? (referral hospital)

Thank you, the text has been edited as suggested

99 – 183

Anesthesiologic management? (anesthesia management)

Thank you, the text has been edited as suggested

2 - 96 -116 - 127- 178 - 184

Pregnant women? (parturients)

Thank you, the text has been edited as suggested

2 –96 – 109 – 113 – 183 – 185- 192- 287- 291- 324- 326 – 328 – 333-335 382-389- 397- 399

The authors capitalized letters of the words in the text where is not necessary. For example, in page 2, Cefazolin, Clindamycin, Cerebro-Spinal Fluid, Gauge Bupivacaine, Epinephrine, ….etc. Meanwhile, the terms of SpO2, ECG, NIBP, MAP, SAP, etc should be spelled out when appeared for the first time in the text.

Thank you, the text has been edited as suggested

 In M&M

Abstract

•           Background: “The objective of the study is to illustrate…” should be stated in the Background, instead of in the Conclusion.

•           Method: Instead of describing Inclusion method, other details should be included.

•           Results: Too simplified.

•           Conclusion: Should make a conclusion which is supported by the solid evidence from study results.

Thank you, the text has been edited as suggested

From 98 to 116

Introduction

Page 1, line 36: “aerosol-generating procedures” appeared in the text for the first time. Therefore, it would be better to describe this term here.

Thank you, the text has been edited as suggested. It is now defined ‘aerosol-generating procedures’

We added as follows:
, i.e. procedures that produce aerosol of respiratory secretions such as: intubation; extubation or related procedures; manual ventilation; open suctioning; non-invasive ventilation.

From 179 to 181

Page 1, line 37: “loco-regional anesthesia”? (local anesthesia and regional anesthesia)

Thank you, the text has been edited as suggested

181

Page 1, line 33, 38, etc: “operators”? (surgeons, airway managers, airway operators, intubators)

Thank you, the text has been edited as suggested

186

Page 1, line 37: “to avoid the management of the patient airway”? (to avoid invasive airway management, e.g., tracheal intubation, on the parturients)

Thank you, the text has been edited as suggested

182

Page 1, line 40: “management protocols in anesthesiology”? (protocols for anesthesia management)

Thank you, the text has been edited as suggested

184 -185

Page 1, line 40: “The objective of this study is to validate….”? (…..the safety and effectiveness of anesthesia management….)

Thank you, the text has been edited as suggested

184

Page 2, line 45: The IRB approval letter was issued on April 3, 2020 and the study began from March 1 to November 30, 2020. So, is this study categorized as prospective or retrospective study?

Thank you, we apologize for the mistake. Patients enrollment began in May 2020, after the Ethics Committee Approval. The text has been edited.

100 - 183

Page 2, line 51: “A total of 152 patients were included…”? Is there any data for the recruitment rate?

Thank you for the interesting remark. Unfortunately, data on the recruitment rate were not available.

-

Page 2, line 57: “All women admitted for inpatient admission..”? (..admitted as an inpatient…)

Thank you, the text has been edited as suggested

204

Page 2, line 58: “Positive patients were placed in a dedicated room, along with those who arrived already with a RT-PCR test.”? (…already with a “positive” RT-PCR test?)

Thank you, the text has been edited as suggested

205

Page 2, line 61: Regarding the role of PPE, the authors cite the reference number 4, which is a scenario-design using standard patients. Any other guidelines papers are available for citing the role of PPE for COVID-19 ?

Thank you, the text has been edited as suggested. We added reference 5

From 436 to 440

Page 2, line 49: There were 152 patients tested positive using RT-PCR and included in this study. In comparison to these 152 COVID-positive obstetrical patients, how many were there COVID-negative obstetrical patients during this study period?

Thank you for the interesting remark. During the study period there were about 2,500 COVID negative patients

289

Page 2, line 64: There were 78+10 (88) subjects received Cesarean section and 44 received vaginal delivery. Is this Cesarean section rate much higher than usual?

Thank you for the interesting remark. Cesarean section rate was higher than usual. In our department CS usual rate is 47%

291

In the Method section, (Page 2 to 3), the authors described all the details of the anesthesia and delivery-related issues. Are all of these details of the procedures for COVID-positive patients different from usual routines and medications (except adopting PPE)?

Thank you for the interesting remark. The management of group 3 and 4 was different compared to covid negative women. . Our anesthetic strategy compared to a control group not affected by SarsCov2 infection changes in the vaginal delivery group and in the spontaneous or induced abortion group. The epidural catheter was positioned before usual in all women although it was not used for childbirth analgesia. We normally performed sedation in spontaneous or induced abortion..

-

Page 2, line 95: Regarding PCEA for the group 1, the authors used ropivacaine without any other narcotics for rescue. Meanwhile, side effects of narcotics (page 3, line 99, nausea, vomiting, pruritus) were monitored every6 hours post-operatively. Any other pain control was used postoperatively?

Thank you for the interesting remark.

Visual Analogic Scale (VAS), presence of adverse effects such as nausea or vomiting or pruritus, the severity and the onset of news symptoms were carefully monitored

From 261 to 264

Page 3, lines 112 and 138: “Sufentanyl” should be spelled as “sufentanil”

Thank you, the text has been edited as suggested

242 - 256

Page 4, line 153, Page 5, line 169: Please define “severe and critical symptoms”

Thank you, the text has been edited as suggested

From 294 to 297

Page 4, line 161: “None of the 152 included women had general anesthesia.” Perhaps it is helpful to discuss later in the Discussion section, if general anesthesia is required in such situation, what would you have done?

Thank you for the interesting remark. Our target has been to perform loco-regional anesthesia in all patients. Opioid free general anesthesia was the contingency plan for those COVID-19 parturients when regional anesthesia and analgesia were not possible.

276-277

Discussion

Regarding the severity and critical conditions of the COVID patients, is there any assessment to better describe 152 included patients in more details (c.f. Table 2)?

Thank you, the table has been edited as suggested

Table 2

 In those more severe or critical conditions of the subjects included in this study, is the treatment course different from those not included in this study in your hospital?

Thank you for the interesting remark. No, no difference

-

Page 6, line 184: Regarding the statement “women are more susceptible to adverse outcome, including the need for endotracheal intubation and admission to an intensive care unit”, any difference between those and your results?

Thank you for the interesting remark. These comparisons were not performed in our study

-

Page 6, line 210: “However, cesarean delivery was used to assist mothers and fetus safely by reducing the coronavirus exposure by operators. Over time, the procedures have improved and optimized.” Please explain what have been improved and optimized?

Thank you for the interesting remark. We agree that this sentence is not clear. Therefore, it has been deleted from the main text.

-

Page 6, line 216: Regarding the issues of “thromboembolism prophylaxis”, “thrombocytopenia”, any data or comparison can be provided in this study?

Thank you for the interesting remark.

Unfortunately no data are available. To avoid any risk of misunderstanding we may delete the sentence.

-

Page 6, line 221: Regarding the issues of “side effects of neuraxial anesthesia such as nausea, vomiting, hypotension, and bradycardia.”, any data can be provided in this stud?

Thank you for the interesting remark. We have added a table with these data.

259 – 260

313

Table 3

Page 7, line 230: Regarding “SpO2 (up to 10% more than the measurement before extraction) and PaO2/FiO2 ratio ….”, in the Methods and Results section, only NIBP was mentioned. Any A-line was implemented or only POC?

Thank you for the interesting remark. Preoperative blood gas in the patients with oxygen therapy was compared to the blood gas post extraction one. We evaluated the improvement of the P/F.

Table 2

307 – 312

376-381

Page 7, line 233: Regarding “Avoid allowing Cerebro-Spinal Fluid (CSF) to drip from the needle and avoid contact with CSF. ”, was there any chance to collect CSF for PCR testing in this study?

Thank you for the interesting remark.

Unfortunately there was not chance to collect CSF for PCR testing in our study

-

Page 7, line 235: “In our experience we planned neuraxial anesthesia for all patients scheduled for elective procedure in order to decrease aerosolization during preoxygenation, face-mask ventilation, endotracheal intubation, oral or tracheal suctioning and extubation.” The emergent Cesarean section for group 2 was elective?

Thank you, the text has been edited.

From 401 to 403

The main issues needed to be discussed or presented

How many and how severe COVID patients are identified in this 152-patient study? Are there any differences from those COVID patient without pregnancy?

Thank you, the text has been edited as suggested.

From 292 to 299

From 307 to 312

From 377 to 381

What is contingency plan for those COVID-parturients when regional anesthesia and analgesia were not possible?

Thank you, the text has been edited as suggested.

276 - 277

While suggesting the neuraxial anesthesia and analgesia is better and safer for COVID-positive parturients during the pandemic, the authors better to provide some indices to support this statement (e.g., safety parameters, efficiency indices, outcomes,..)

Thank you, the text has been edited as suggested.

From 293 to 300

Table 3

319 -320

Table 4

Round 2

Reviewer 1 Report

Thank you so much for improving the manuscript by addressing each question.

In this new version the manuscript has been improved and runs very well.
Finally, the treatment strategy and conclusions are understood.

Author Response

We thanks the reviewer for the positive comments

Reviewer 3 Report

Manuscript Number: healthcare-1574741

Authors: Coviello et al.

Title of the manuscript: Protocols of Anesthesiologic Management ….

20220223

The authors did a diligent work in revising the manuscript according to reviewer’s comments. Here, I suggest some issues which need to be taken care before accepting it to be published. Mainly, I suggest to carefully deal with the statistical design and analysis (including the presentation in the Tables).

Major comments:

  • Page 11, line 267: The authors added the primary and secondary outcomes in the revised text.
    • Page 11, line 268: “patients’ safety and effectiveness minimizing aerosol-generating procedures.”??? Should be “patients’ safety and effectiveness peri-operatively”?? Please explain the observing parameters for the safety (e.g., vital signs?) and effectiveness (anesthesia quality?)
    • Page 11, line 271: “providers safety minimizing aerosol-generating procedures.”?? Should be “providers’ safety perioperatively”?? Again, it should be indicated how to measure the safety (the symptoms or RT-PCR test results?)
  • Page 11, line 278: The authors added the statement of SAP. It is better to explain which part of study results applied to Chi-square test and which to Student’s t test. Such statement should be included in the annotation of each Table.
  • Page 13, line 310: About the statement of Table 2, please indicate which two groups were in comparison and reached a statistically significant difference. Or simply shows the incidence rate.
  • Page 13, line 313: The statement of Table 3 in the text is not consistent the contents of the Table 3 (e.g., p < 0.05 ??? for which comparison). Or simply shows the incidence rate.
  • Page 13, line 319: Similarly, the statement about Table 4 is not consistent with the one of the Table 4 (p < 0.05 for which comparison?)
  •  

Minor comments:

  • Page 4, line 104: “Sars-Cov2” should be “SARS-Cov-2”
  • Page 4, line 109: “None of the included women….., an increase of the severity symptoms or onset…”??? Should be “…no increase of the ….”?
  • Page 11, line 275: “General anesthesia opioid free” should be “Opioid-free general anesthesia”
  • Page 12, line 294: “present symptoms” should be “presenting symptoms”
  • Page 12, line 297: “appendix 1”??
  • Page 23, line 543-545: The contents are repeated and redundant.
  • Page 17, line 403: “effectiveness” should be “effective”

Author Response

Healthcare-1574741

Protocols of Anesthesiologic Management in pregnant women with SARS-COV-2 infection

Dear Editor,

thank you for the time spent in reviewing our manuscript and for the opportunity to re-revise it. We have carefully considered all the comments by reviewers and we have addressed them in a point-by-point way. Please, find a table below with a list of our replies and the changes applied to the paper. We do believe that it’s now much improved and we hope you will find it suitable for publication in your journal.

We look forward to hearing from you.

Best regards

The authors

Comment

Reply

Line

Reviewer #3               

The authors did a diligent work in revising the manuscript according to reviewer’s comments. Here, I suggest some issues which need to be taken care before accepting it to be published. Mainly, I suggest to carefully deal with the statistical design and analysis (including the presentation in the Tables).

Major comments:

•          Page 11, line 267: The authors added the primary and secondary outcomes in the revised text.

•  Page 11, line 268: “patients’ safety and effectiveness minimizing aerosol-generating procedures.”??? Should be “patients’ safety and effectiveness peri-operatively”?? Please explain the observing parameters for the safety (e.g., vital signs?) and effectiveness (anesthesia quality?)

Thank you, the text has been edited as suggested

266– 267

• Page 11, line 271: “providers safety minimizing aerosol-generating procedures.”?? Should be “providers’ safety perioperatively”?? Again, it should be indicated how to measure the safety (the symptoms or RT-PCR test results?)

Thank you, the text has been edited as suggested.

270 – 271

•          Page 11, line 278: The authors added the statement of SAP. It is better to explain which part of study results applied to Chi-square test and which to Student’s t test. Such statement should be included in the annotation of each Table.

Thank you for the interesting remark.  As described in the statistical analysis section, chi-square test was used for categorical variables, while T test was used for continuous variables.

280 – 282

•  Page 13, line 310: About the statement of Table 2, please indicate which two groups were in comparison and reached a statistically significant difference. Or simply shows the incidence rate.

Thank you, the table has been edited as suggested.

Table 2

310- 313

•          Page 13, line 313: The statement of Table 3 in the text is not consistent the contents of the Table 3 (e.g., p < 0.05 ??? for which comparison). Or simply shows the incidence rate.

Thank you, the table has been edited as suggested. We are sorry for the typo. P value of Table 3 was not deleted.

Table 3

315 – 316

•  Page 13, line 319: Similarly, the statement about Table 4 is not consistent with the one of the Table 4 (p < 0.05 for which comparison?)

Thank you, the table has been edited as suggested. We are sorry for the typo. P value of Table 4 was not deleted.

Table 4

Minor comments:

•        Page 4, line 104: “Sars-Cov2” should be “SARS-Cov-2”

Thank you, the text has been edited as suggested.

104

•     Page 4, line 109: “None of the included women….., an increase of the severity symptoms or onset…”??? Should be “…no increase of the ….”?

Thank you, the text has been edited as suggested.

110

•   Page 11, line 275: “General anesthesia opioid free” should be “Opioid-free general anesthesia”

Thank you, the text has been edited as suggested.

275

•   Page 12, line 294: “present symptoms” should be “presenting symptoms”

Thank you, the text has been edited as suggested.

295

• Page 12, line 297: “appendix 1”??

Thank you, the table has been edited as suggested.

289

•   Page 23, line 543-545: The contents are repeated and redundant.

Thank you, the table has been edited as suggested.

Table 2

•           Page 17, line 403: “effectiveness” should be “effective”

Thank you, the text has been edited as suggested.

405
